# Node Deployment Optimization for Wireless Sensor Networks Based on Virtual Force-Directed Particle Swarm Optimization Algorithm and Evidence Theory

**DOI:** 10.3390/e24111637

**Published:** 2022-11-10

**Authors:** Liangshun Wu, Junsuo Qu, Haonan Shi, Pengfei Li

**Affiliations:** 1Xi’an Key Laboratory of Advanced Control and Intelligent Processing, School of Automation, Xi’an University of Posts and Telecommunications, Xi’an 710061, China; 2School of Electronic Information and Electrical Engineering, Shanghai Jiao Tong University, Shanghai 200241, China; 3Xi’an Robertic Intelligent Systems International Science and Technology Cooperation Base, Xi’an 710061, China

**Keywords:** wireless sensor network, D-S evidence, virtual force, particle swarm optimization

## Abstract

Wireless sensor network deployment should be optimized to maximize network coverage. The D-S evidence theory is an effective means of information fusion that can handle not only uncertainty and inconsistency, but also ambiguity and instability. This work develops a node sensing probability model based on D-S evidence. When there are major evidence disputes, the priority factor is introduced to reassign the sensing probability, with the purpose of addressing the issue of the traditional D-S evidence theory aggregation rule not conforming to the actual scenario and producing an erroneous result. For optimizing node deployment, a virtual force-directed particle swarm optimization approach is proposed, and the optimization goal is to maximize network coverage. The approach employs the virtual force algorithm, whose virtual forces are fine-tuned by the sensing probability. The sensing probability is fused by D-S evidence to drive particle swarm evolution and accelerate convergence. The simulation results show that the virtual force-directed particle swarm optimization approach improves network coverage while taking less time.

## 1. Introduction

Wireless sensor networks (WSNs) are commonly employed in harsh, high-risk, and distant unmanned environments. The sensor nodes in WSNs installed underwater (oceans, shallow seas, and lakes) are sometimes difficult to identify, and often swing or drift with the influence of ocean currents, waves, weather, ships, fish, and other factors. Physical causes of displacement or failure also have an impact on underwater acoustic detection equipment. Furthermore, monitoring signals attenuate quickly in an underwater environment, and different types of noise interference might occur often, resulting in quite significant energy loss of sensor nodes. As a result, while deploying wireless sensor network nodes, it is important to consider not only network coverage, but also the node displacement distance.

Many heuristic techniques, such as the genetic algorithm [1], the fish optimization algorithm [2], the ant colony algorithm [3], and the particle swarm optimization algorithm [4], are employed to optimize the placement of sensor nodes. These algorithms necessitate several repetitive computations. Wang et al. [5] suggested a virtual force-directed particle swarm algorithm (VF-PSO) deployment approach in which the moving distance is dictated by the connection between nodes, which is highly influenced by node density. Here, virtual force (VF) refers to the degree of mutual interference measured by the distance between nodes.

In this paper, a novel node deployment optimization strategy for WSNs is suggested. The virtual force-directed particle swarm (VF-PSO) algorithm and the D-S evidence theory are employed in the strategy. We provide the following contributions:A sensing probabilistic model is constructed. To aggregate the sensing probability in the circular area with radius *R* around the sensor node, the D-S evidence theory is applied. When there is a considerable evidence conflict, the typical D-S evidence theory aggregation rule does not conform to the real circumstances, and the result is not convincing. To address the concern, the priority factor is introduced to reassign the sensing probability.A virtual force-directed particle swarm optimization algorithm with sensing probabilities aggregated by D-S evidence (VF-PSO-DS for short) is proposed. The approach improves the virtual force-directed particle swarm optimization (VF-PSO) algorithm by considering sensing probability. The sensing probability is aggregated by D-S evidence, which guides the particle swarm evolution process and accelerates convergence.

The rest of this paper is organized as follows. Section 2 reviews related work. Section 3 presents and improves the D-S evidence theory. Section 4 presents the construction of the system model. Section 5 proposes the improved virtual force-directed particle swarm optimization algorithm (VF-PSO-DS). Section 6 presents the simulation and explains the results. The conclusion is reached in Section 7.

## 2. Related Work

Recently, deployment optimization in WSNs has been discussed by many researchers. Most of them consider this problem under different constraints and deployment goals. Dou et al. [6], for example, look at the optimization of sensing coverage and regional connectivity in both deterministic and random deployment. Hosseinzadeh et al. [7] offered a multi-agent distributed solution for WSN deployment. The optimization is considered a linear programming problem with time-invariant box constraints and potentially time-varying inequality constraints. Kang [8] discovered that the average number of clusters and the transmission range of control messages have a substantial influence on network lifetime. Liu et al. [9] present an optimization approach for WSN node deployment based on an ant lion optimization algorithm to address difficulties in WSN node deployment such as uneven distribution of nodes and incomplete coverage. Other influential works include [7,8,9,10,11]. Table 1 lists the above-mentioned work by constraints.

The virtual force (VF) approach has been a prominent WSN node deployment strategy for the past 15 years. In 2007, Wang et al. [5] suggested a virtual force-directed PSO (VF-PSO) algorithm for WSN deployment in an early work in this area. Yu et al. [15] proposed a self-deployment algorithm for nodes with mobility based on virtual force (VF). Xiaoping et al. [16] investigated the characteristics of four VF models and then assessed these models with several indicators: coverage increment scale, iterative number, coverage efficiency, and the average movement distance of nodes. They proved that the VF model with three parameters was superior, as was the VF model that does not include VF value but rather direction. To solve the problems of connectivity maintenance and node stacking in VF models, an extended VF-based approach was studied to achieve optimal deployment [17]. Yu et al. [18] proposed a sensor deployment strategy for mobile WSNs based on the van der Waals force. It is critical to improve sensing coverage and network connectivity to provide reliable communication [19]; therefore, a hybrid local virtual force algorithm (HLVFA) that incorporates the constraints of sensing coverage and network connectivity was proposed [20]. Theoretical analysis has shown that a hexagonal structure is the best two-dimensional network for providing the greatest coverage area with the fewest sensor nodes and the least amount of energy usage [21]. Yu et al. [22] reported a novel VF model based on virtual spring force (VFA-SF) and examined the associated efficiency in detail using statistical analysis. To place the self-consistent nodes in large-scale WSNs, a hybrid optimization algorithm based on two separate VF algorithms inspired by the interactions among physical sensor nodes was presented in [23]. Afterward, a method based on a 3D VF model driven by self-adaptive deployment (named 3DVFSD) was proposed in [24].

The application of D-S evidence theory in node deployment for WSNs has long been a hot topic. Zhao et al. [25] sought to reduce the influence of incomplete original or subjective parameters in evaluating the reliability of a distribution system. For WSNs, many factors, such as mutual interference of wireless links, battlefield applications, and nodes exposed to the environment without good physical protection, result in the sensor nodes being more vulnerable to attacks. To ensure network security, NBBTE (banding belief theory of node behavioral strategies of the trust evaluation algorithm) has been developed, which integrates the approach of node behavioral strategies with evidence theory [26]. Miao et al. [27] proposed a trustworthiness evaluation method for sensor nodes that can perceive multi-dimensional data based on D-S evidence theory at the data level. The Zigbee network was used to monitor the environmental parameters of aquiculture water [28]. Because artificial surveys have low accuracy, a gas pipeline leakage diagnosis system based on BP neural networks and D-S theory was presented by introducing WSN and information fusion theory [29]. Zhao et al. [30] proposed a novel fleet deployment approach based on route risk evaluation to fully use navigation resources while reducing risks. Song et al. [31] investigated many evidence theory-based approaches for node deployment optimization in WSNs. To improve the security in WSNs, Sun et. al. [32] proposed a trust ant colony routing algorithm by including a node trust evaluation model based on D-S evidence theory into the ant colony routing protocol. Song et. al. [33] presented a novel node deployment scheme that was based on the evidence theory approach and catered to 3D USWNs. To eliminate dependence on network training, an improved D-S evidence theory was applied to the host assessment [34].

## 3. D-S Evidence Theory with Priority Factors

### 3.1. D-S Evidence Theory 

The Dempster–Shafer evidence theory, often known as the D-S evidence theory, is a widely used technique in the field of multi-sensor information fusion. D-S evidence theory, which is an inexact derivation of probability theory and is based on Bayesian reasoning, can work without a priori information and conditional probability [35]. D-S evidence theory begins with the assumption of a complete and incompatible element set (also called a recognition framework), denoted by Θ. Any element in the recognition framework is represented by $\Theta =\left\{H_{0},H_{1}\right\}$, where $H_{0}$ means “was not perceived by the sensor node” and $H_{1}$ means “was perceived by sensor node”.

Let $m_{1}$ and $m_{2}$ be the basic probability distribution functions in the same recognition framework Θ, with focal elements $A_{i}\left(i=1,2,\cdots ,k\right)$ and $B_{i}\left(i=1,2,\cdots ,k\right)$, and then the aggregation formula of the two pieces of evidence is
(1)$$m_{1}⨁m_{2}\left(A\right)=\{\begin{array}{l} 0,A=\emptyset \\ \frac{\sum_{A_{i}\cap B_{i}}m_{1}\left(A_{i}\right)m_{2}\left(B_{i}\right)}{1-K},A\neq \emptyset^{\prime } \end{array}$$
where $K=\sum_{A_{i}\cap B_{i}=\emptyset }m_{1}\left(A_{i}\right)m_{2}\left(B_{i}\right)$, which indicates the degree of evidence conflict. ⨁ denotes exclusive-or calculation.

The aggregation formula of N pieces of evidence is
(2)$$m\left(A\right)=m_{1}⨁m_{2}⨁m_{3}\cdots ⨁m_{N}\left(A\right)$$
Then, a basic probability function of joint action of multiple pieces of evidence is obtained.

### 3.2. The Problem with D-S Evidence Theory

A basic probability assignment function of *n* pieces of evidence is
(3)$$m_{1}=\{\begin{array}{c} m\left(\Phi \right)=0\\ m\left(E\right)=1\\ m\left(O\right)=0\\ m\left(U\right)=0 \end{array}$$
(4)$$m_{n}=m_{n-1}=m_{n-2}=\cdots m_{2}=\{\begin{array}{l} m\left(\Phi \right)=0\\ m\left(E\right)=k_{1},0\leq k_{1}<1\\ m\left(O\right)=k_{2},0\leq k_{2}<1\\ m\left(U\right)=1-k_{1}-k_{2},0\leq 1-k_{1}-k_{2}<1 \end{array}$$

It can be seen from Formula (4) that if $k_{2}=1$, there is a conflict between the first piece of evidence and the (n−1)-th piece of evidence. When calculating the basic probability assignment after fusion, the denominator is 0, so the following calculation cannot be carried out normally, so the general evidence theory cannot easily solve this problem.

If $k_{2}\neq 1$, the following can be obtained:(5)$$m=m_{1}⊕m_{2}⊕\ldots ⊕m_{n}=\{\begin{array}{c} m_{1}\left(\Phi \right)=0\\ m_{1}\left(E\right)=1\\ m_{1}\left(O\right)=0\\ m_{1}\left(U\right)=0 \end{array}=m_{1},$$

According to Equation (5), the other n−1 pieces of evidence have nothing to do with the result of information fusion, but they are the same as the first one. This shows that all the n−1 pieces of evidence in this evidence are the same (i.e., $m_{2}$), and this shows that the result of information fusion has nothing to do with the n−1 pieces of evidence. The first piece of evidence denies the n−1 pieces of evidence, which is wrong.

If there is a conflict between one piece of evidence and other pieces of evidence, but the result of data fusion can be expressed, the denominator of zero will appear in the aggregation rules, resulting in the inability to use the D-S aggregation rules. Therefore, we should avoid the inconsistent results of the D-S evidence theory when it conflicts and synthesizes multiple pieces of evidence.

### 3.3. Introduction of Priority Factors

The priority factor is usually a correction method used in system decision-making. In this paper, the priority factor method is introduced into evidence aggregation rules, and the evidence provided by information sources is divided according to the priority level, to further solve the problems of invalidation and conflict in the D-S evidence theory aggregation rules.

#### 3.3.1. Definition of Priority Factors

If there are n objective factors $G_{1},G_{2},\ldots G_{n}$ that need to be considered when making a decision, these factors may be too many and inconsistent to be judged directly, and the priority factor can be used to solve this problem. First, we need to compare these objective factors, such as objective factors $G_{i}$ and $G_{j}$. If $G_{i}$ is considered more important, it will be recorded as $G_{i}>G_{j}$; otherwise, it will be recorded as $G_{i}<G_{j}$. Then, we count the times that $G_{i}$ appear on the left side of the inequality symbol and arrange the target factors in descending order according to these times so that the priority order of the target factors can be obtained. Finally, the system will make decisions according to this order.

Considering that different sources of information provide different priority levels of evidence, the priority factor $P_{i}$ (*i* = 1,2,…*L*) is defined to indicate the priority level of each piece of evidence. The priority factor is just a symbol. Whenever a piece of evidence is listed as the most important piece of evidence, the priority factor is assigned to $P_{1}$. If the priority factor level of a piece of evidence is in the k-th place, the priority factor is $P_{k}$. A priority factor can also be regarded as a special normal number and a weight coefficient with special meaning as evidence. The relationship between each priority factor $P_{i}$ (*i* = 1,2,…*L*) is $P_{1}>P_{2}>\ldots >P_{L}$. If any positive number M, $P_{i}\gg MP_{i+1}$ can be established, it is said that $P_{i}$ is far greater than $P_{i+1}$, marked as $P_{i}\gg P_{i+1}$, which indicates that $P_{i}$ and $P_{i+1}$ are not level quantities.

#### 3.3.2. Determination of Priority Factors

To determine the priority factor of evidence, it is necessary to define the amount of self-conflict in the evidence. The amount of self-conflict of the evidence is the sum of conflicts between evidence i and evidence j (*j* = 1,2,…*i* − 1,*i* + 1,…, *n*), and its relationship is as follows:(6)$$\varphi_{i}=\sum_{j=1,j\neq i}^{n}\sum_{A_{i}\cap A_{j}=\Phi }m_{i}\left(A_{i}\right)m_{j}\left(A_{j}\right),$$
where n represents the amount of evidence.

The steps to determine the priority factor are as follows: First, all evidence pieces should be collected, and the self-conflict amount $\varphi_{i}$ (*i* = 1,2,…n) of each piece of evidence should be processed and calculated, and then the self-conflict amount $\varphi_{i}$ should be sorted from the largest to the smallest. The priority factor $P_{1}$ is the piece of evidence with the smallest conflict amount, and the priority factor $P_{L}$ refers to the piece of evidence with the largest self-conflict amount. The divergence of some evidence usually leads to conflicts between two pieces of evidence, so these conflicting pieces of evidence should have a relatively large number of self-conflicts $\varphi_{i}$. If there are two pieces of evidence with the same number of self-conflicts $\varphi_{i}$, then the values of the priority factors of the two pieces will be the same. The calculated priority factors are arranged according to the size of the last subscript, which is convenient for the subsequent synthesis.

### 3.4. D-S Evidence Aggregation Rules with Priority Factors

According to the above definition, the priority factor method of evidence synthesis is introduced. Firstly, the information about each piece of evidence is collected, and then the self-conflict quantity of evidence is calculated. According to the self-conflict quantity, the priority factor $P_{i}$ (*i* = 1,2,…*L*) is determined, and the value range of L is n≥L≥1. By fusing the pieces of evidence of these priority factors $P_{k}$, the k value is k=1,2,…L, when only the only evidence is left, the priority factor $P_{k}$ will not need to be fused continuously, and the number of existing pieces of evidence is L.

When the number of pieces of evidence L satisfies L≥2, it means that the conflicts have occurred in the fusion of pieces of evidence, and the conflicts need to be dealt with. By using the aggregation rule of D-S evidence theory, the evidence with less conflict is directly fused, and the aggregation method of evidence with a priority factor is used to deal with the evidence with the highest conflict, thus reducing the conflict between two pieces of evidence.

This method uses conflicting evidence m(A) to assign values to different propositions, so the assignment relationship is
(7)$$\{\begin{array}{l} m\left(\Phi \right)=0\\ m\left(A\right)=\sum_{A_{i}\cap B_{j}\cap \ldots =A}m_{1}\left(A_{i}\right)\cdot m_{2}\left(B_{j}\right)\ldots +f\left(A\right),\forall A\neq \Phi \end{array},$$

In Equation (7), the basic probability assignment of conflict evidence is expressed in *f*(*A*), and
(8)$$\{\begin{array}{l} f\left(A\right)\geq 0,f\left(\Phi \right)=0\\ \sum_{A\subset \Theta }f\left(A\right)=k \end{array},$$
where $k=\sum_{A_{i}\cap B_{j}\cap \ldots =\Phi }m_{1}\left(A_{i}\right)\cdot m_{2}\left(B_{j}\right)\ldots$.

By using a voting method, the basic probability assignment with highly conflicting evidence is weighted, so f(A) can be taken as follows:(9)$$\{\begin{array}{l} f\left(A\right)=k\cdot q\left(A\right)\\ q\left(A\right)=\frac{n_{A}}{n} \end{array},$$
where $n_{A}$ is given by
(10)$$n_{A}=\{\begin{array}{l} n_{A}+1,m_{i}\left(A\right)-m_{th}\geq 0\\ n_{A},\;\mathrm{otherwise} \end{array},$$

In Equation (8), all supported propositions are counted. If the given threshold $m_{th}$ is less than or equal to its basic probability assignment, then 1 is added to the value of $n_{A}$, and the statistical total of different propositions is denoted as *n*, while the initial value of the threshold $m_{th}$ is assigned to 1/2.

According to the synthesis method determined above, the method for the synthesis of new evidence again is as follows:

Step 1. The two pieces of evidence of priority factors $P_{1}$ and $P_{2}$ are $E_{1}$ and $E_{2}$. If the amount of mutual conflict $k_{ij}$ is the conflict sum between evidence i and evidence j, then the relationship between them can be expressed as:(11)$$k_{ij}=\sum_{A_{i}\cap B_{j}\cap \ldots =A}m_{i}\left(A_{i}\right)\cdot m_{j}\left(A_{j}\right),$$
where i=1,2,…n and j=1,2,…n. Equation (12) shows the relationship between the amount of mutual conflict $k_{ij}$ and the amount of self-conflict $\varphi_{i}$:(12)$$\varphi_{i}=\sum_{j=1,j\neq i}^{n}k_{ij},$$

Step 2. Calculate the conflict value between them.
(13)$$\sum_{A_{i}\cap A_{j}=\Phi }\prod_{i=1}^{n}m_{i}\left(E_{i}\right),n=2,$$

The amount of mutual conflict calculated by the evidence $E_{1}$ and the evidence $E_{2}$ is $k_{12}$, and then $k_{th}$ is assumed as the conflict threshold. If the relationship between them is $k_{12}<k_{th}$, the aggregation rule of D-S evidence theory will be used to deal with it when synthesizing the evidence. If the relationship between them is $k_{12}\geq k_{th}$, when the evidence is aggregated, the priority factor $P_{1}$ and priority factor $P_{2}$ are added for processing, and then the formula is $m=P_{1}m_{1}\left(E_{1}\right)⊕P_{2}m_{2}\left(E_{2}\right)$.

Therefore, the method of introducing the priority factor is used in evidence synthesis, which is shown as
(14)$$m\left(A\right)=\left[P_{1}m_{1}⊕P_{2}m_{2}\right]\left(A\right)=\{\begin{array}{c} \frac{\sum_{X\cap Y=A}m\left(X\right)\cdot m\left(Y\right)}{1-\sum_{X\cap Y=\Phi }m\left(X\right)\cdot m\left(Y\right)},k_{ij}<k_{th}\\ \sum_{X\cap Y=A}m\left(X\right)\cdot m\left(Y\right)+f\left(A\right),k_{ij}\geq k_{th} \end{array},$$

Step 3. Take the next piece of evidence $E_{3}$ of the next priority factor $P_{3}$, and then re-synthesize the priority factor with the result obtained in the previous step according to the synthesis rules in Step 2.

Step 4. Use the above method to continuously synthesize the newly generated priority factor. When the priority factor is $P_{L}$, the improved result is obtained through fusion calculation. Finally, a reasonable judgment condition is established, and the result is decided.

The synthesis method based on the evidence classification strategy is the evidence synthesis method with a priority factor, and its application process is shown in Figure 1.

## 4. System Model

### 4.1. Sensing Probability Model

If the two-dimensional plane monitoring area Λ is discretized into m×n pixels, we define the event $r_{ij}$ that the i-th pixel $p_{i}$ is sensed by the j-th sensor $s_{j}$, and the probability of this event $P\left\{r_{ij}\right\}=P\left(p_{i},s_{j}\right)$,
(15)$$P\left(p_{i},s_{j}\right)=\{\begin{array}{l} 1,\;if\;d\left(p_{i},s_{j}\right)\leq R_{s}-R_{e}\\ e^{-\lambda {\left(d\left(p_{i},s_{j}\right)-R_{s}+R_{e}\right)}^{\beta }},if\;R_{s}-R_{e}<d\left(p_{i},s_{j}\right)\leq R_{s}+R_{e}\\ 0,\;if\;d\left(p_{i},s_{j}\right)>R_{s}+R_{e} \end{array},$$
where $d\left(p_{i},s_{j}\right)$ is the distance between the i-th pixel $p_{i}$ and the j-th sensor $s_{j}$, λ and β are parameters related to the sensing probability, and $R_{e}$ is the sensing error range of the sensor node $s_{j}$.

By introducing the D-S evidence theory of the priority factor to synthesize perception probability, the probability that this pixel is perceived by a wireless sensor network is as follows:(16)$$P\left(p_{i}\right)=P\left(p_{i},1\right)⨁P\left(p_{i},2\right)⨁P\left(p_{i},3\right)⨁\cdots ⨁P\left(p_{i},N\right)$$

The evidence is aggregated only if the sensing probability $P\left(p_{i},s_{j}\right)>0$.

### 4.2. Area Coverage Model

There are m×n pixels in the monitoring area of the bitplane Λ, and the area of each pixel is expressed as ∆x×∆y (assuming that the area of each pixel is 1 in this paper). The probability of the i-th pixel being sensed by the wireless sensor network is $P\left(p_{i}\right)$; when $P\left(p_{i}\right)\geq P_{\mu }$ ($P_{\mu }$ denotes the lowest sensing probability allowed by the wireless sensor network), this pixel can be regarded as being sensed by the wireless sensor network.

Whether the i-th pixel is sensed by the sensor node is marked as $P_{COV}\left(p_{i}\right)$; i.e.,
(17)$$P_{COV}\left(p_{i}\right)=\{\begin{array}{l} 0,\;\mathrm{if}\;P\left(p_{i}\right)<P_{\mu }\\ 1,\;\mathrm{if}\;P\left(p_{i}\right)\geq P_{\mu } \end{array},$$

In this paper, the coverage of the monitoring area $R_{area}$ is defined as the ratio of the perceived pixel area $\Lambda_{area}$ to the total area of the monitoring area $\Lambda_{s}$, namely
(18)$$R_{area}=\frac{\Lambda_{area}}{\Lambda_{s}}=\frac{\sum_{x=1}^{m}\sum_{y=1}^{n}P_{COV}\left(p_{i}\right)}{m\times n}$$

## 5. Deployment Optimization Using Virtual Force-Directed Particle Swarm Algorithm

### 5.1. Virtual Force

By calculating the sensing probability of pixels in a circular area with the sensor node $s_{j}$ as the center and R as the radius, the movement of the sensor node $s_{j}$ is determined.

Assuming any pixel $p_{i}$ and sensor node $s_{j}$, the force exerted by the sensor node $s_{j}$ by the pixel $p_{i}$ can be expressed as follows:(19)$$F\left(p_{i},s_{j}\right)=\{\begin{array}{l} \alpha \frac{P\left(i\right)-k}{d\left(p_{i},s_{j}\right)},0<d\left(p_{i},s_{j}\right)<R\\ 0,d\left(p_{i},s_{j}\right)\geq R \end{array}.$$
where P(i) is the comprehensive probability that the i-th pixel $p_{i}$ is perceived by the wireless sensor network, k and α denote the gain coefficients, and $d\left(p_{i},s_{j}\right)$ is the distance between the pixel $p_{i}$ and the sensor node $s_{j}$.

The force direction points from the pixel $p_{i}$ to the sensor node $s_{j}$, and the component of the X-axis $F_{x}\left(p_{i},s_{j}\right)$ and the component of the Y-axis $F_{y}\left(p_{i},s_{j}\right)$ can be obtained after decomposition.

The resultant force in the X-axis direction is $F_{x}\left(s_{j}\right)=\sum F_{x}\left(p_{i},s_{j}\right)$, and the resultant force in the Y-axis direction is $F_{y}\left(s_{j}\right)=\sum F_{y}\left(p_{i},s_{j}\right)s_{j}$, so the resultant force generated by the pixels in the circular area with radius R on the sensor node is $F_{xy}\left(s_{j}\right)=\sqrt{F_{x}^{2}\left(s_{j}\right)+F_{y}^{2}\left(s_{j}\right)}$.

### 5.2. Optimization Problem Formulation

The goal of deployment optimization is to maximize aware coverage
(20)$$\max\left\{R_{area}\right\}=\max\left\{\frac{\Lambda_{area}}{\Lambda_{s}}\right\}$$
where $R_{area}=\frac{\Lambda_{area}}{\Lambda_{s}}$ can be obtained by the following equations:
(21)$$\left\{ \begin{array}{l} \frac{\Lambda_{area}}{\Lambda_{s}}=\frac{\sum_{x=1}^{m}\sum_{y=1}^{n}P_{COV}\left(p_{i}\right)}{m\times n}\\ P_{COV}\left(p_{i}\right)=\{\begin{array}{l} 0,\mathrm{if}P\left(p_{i}\right)<P_{u}\\ 1,\mathrm{if}P\left(p_{i}\right)\geq P_{u} \end{array}\\ P\left(p_{i}\right)=P\left(p_{i},1\right)⨁P\left(p_{i},2\right)⨁P\left(p_{i},3\right)⨁\cdots ⨁P\left(p_{i},N\right)\\ P\left(p_{i},s_{j}\right)=\{\begin{array}{l} 1,if\;d\left(p_{i},s_{j}\right)\leq R_{s}-R_{e}\\ e^{{-\lambda (d(p_{i},s_{j})-R_{s}+R_{e})}^{\beta }},if\;R_{s}-R_{e}<d(p_{i},s_{j})\leq R_{s}+R_{e}\\ 0,if\;d\left(p_{i},s_{j}\right)>R_{s}+R_{e} \end{array}\\ F\left(p_{i},s_{j}\right)=\{\begin{array}{l} \alpha \frac{P(i)-k}{d(p_{i},s_{j})},0<d(p_{i},s_{j})<R\\ 0,d(p_{i},s_{j})\geq R \end{array}\\ F_{x}(s_{j})=\sum F_{x}(p_{i},s_{j})\\ F_{y}(s_{j})=\sum F_{y}(p_{i},s_{j})\\ F_{xy}(s_{j})=\sqrt{F_{x}^{2}(s_{j})+F_{y}^{2}(s_{j})}\\ F_{x}\left(p_{i},s_{j}\right)=F(p_{i},s_{j})sin\theta \\ F_{y}\left(p_{i},s_{j}\right)=F(p_{i},s_{j})cos\theta \end{array}\right.$$
where θ measures the angle between the sensor $s_{j}$ and the pixel $p_{i}$.

The constraints are listed as follows:(22)$$s.t.\{\begin{array}{l} 0<d\left(p_{i},s_{j}\right)<R\\ d\left(p_{i},s_{j}\right)=\sqrt{d_{x}^{2}\left(p_{i},s_{j}\right)+d_{y}^{2}\left(p_{i},s_{j}\right)}\\ 0<d_{x}\left(o,s_{j}\right)<R\\ 0<d_{y}\left(o,s_{j}\right)<R \end{array}$$

It can be seen from (21) that $d\left(p_{i},s_{j}\right)$ (i.e., the distance between the sensor $s_{j}$ and the pixel $p_{i}$) is the variable need to be determined. From (22), we can say that the optimization problem is, in fact, adjusting the locations of the sensor set $\left\{s_{j}\right\}$. What needs to be done is to adjust the lateral movement distance $d_{x}\left(o,s_{j}\right)$ and longitudinal movement distance $d_{y}\left(o,s_{j}\right)$ to the central point o of the circular area with radius R, so as to maximize the coverage.

The distribution of pixel set {$p_{i}$} is known. For an example, see Figure 2.

### 5.3. The Process of Particle Swarm Optimization

Particle swarm optimization (PSO) regards each sensor node as a massless particle flying at a certain speed in the search space. This speed is dynamically adjusted according to the flight experience of itself and its companions. Reference [10] discusses in detail that the optimal position and velocity of the node particles in particle swarm optimization depend on the initial random position and velocity of the nodes and the evolution formula. Because the virtual force algorithm can effectively guide the distribution process of mobile nodes, and the swarm intelligence optimization strategy has a strong global optimization ability, this paper adds the influence of virtual force into the evolution formula of the swarm intelligence algorithm, and its evolution formula is as follows:(23)$$v_{q,Dim}\left(t+1\right)=w\left(t\right)\times v_{q,Dim}\left(t\right)+c_{1}\mathrm{Rand}()\left(L_{q,Dim}^{*}\left(t\right)-x_{q,Dim}\left(t\right)\right)+\;c_{2}\mathrm{Rand}()\left(L_{Dim}^{*}\left(t\right)-L_{q,Dim}\left(t\right)+c_{3}\mathrm{Rand}()v_{q,Dim}\right)$$
where the particle swarm size is Q; $v_{q;Dim}\left(t\right)$ is the velocity of the particle q in the Dim-th dimension in the t-th generation; Dim=1 expresses the transverse direction (X-axis); Dim=2 expresses the longitudinal direction (Y-axis); w(t) is inertia weight; $c_{1}$ and $c_{2}$ are acceleration constants; $c_{3}$ is the acceleration factor used to adjust the influence of virtual force; and Rand() denotes the random function varying in the range of [0, 1], which should gradually decrease with the increase of iteration times. $L_{q,Dim}^{*}$ is the best position of the particle q in the Dim-th dimension, the index number of the best position of all particles in the Dim-th dimension is $L_{Dim}^{*}$, and the current position of the particle q in the Dim-th dimension is $L_{q,Dim}$. We have
(24)$$w\left(t\right)=0.9-\frac{t}{MaxNumber}\times 0.5.$$
where MaxNumber is the maximum number of iterations and t denotes the number of iterations. Driven by virtual force, the lateral movement distance $∆x_{q}\left(t\right)$ of the particle q in the t-th generation is
(25)$$∆x_{q}\left(t\right)=\{\begin{array}{l} 0,\left|F_{xy}\right|\leq F_{th}\\ \frac{F_{x}^{q}}{F_{xy}^{q}}\times Max\mathrm{Step}\times e^{-F_{xy}^{-1}},\left|F_{xy}\right|>F_{th} \end{array},Dim=1.$$

The longitudinal moving distance $∆y_{q}\left(t\right)$ of the particle q in the t-th generation is
(26)$$∆y_{q}\left(t\right)=\{\begin{array}{l} 0,\left|F_{xy}\right|\leq F_{th}\\ \frac{F_{y}^{q}}{F_{xy}^{q}}\times Max\mathrm{Step}\times e^{-F_{xy}^{-1}},\left|F_{xy}\right|>F_{th} \end{array},Dim=2.$$
where MaxStep is the maximum allowable moving distance of the sensor node, $F_{xy}^{q}$ is the virtual force acting on the particle q, $F_{x}^{q}$ and $F_{y}^{q}$ are the virtual force components of the particle q on the X-axis and the Y-axis, and $F_{th}$ is the threshold value of the virtual force.

### 5.4. Deployment Strategy

The algorithm steps are as follows:

Step 1. Each wireless sensor node in the area Λ broadcasts information, which includes the index and location information of the node, and enters step 2.

Step 2. If the wireless sensor node $s_{j}$ receives the broadcast information of the neighbor node, it updates the neighbor list information and enters step 3.

Step 3. By introducing the D-S evidence theory of the priority factor, the sensing probability $P\left(p_{i}\right)$ of the pixel $p_{i}$ in the circular area $R=\left(R_{s}+R_{e}\right)$ with a radius around the sensor node $s_{j}$ is aggregated.

Step 4. Using the particle swarm optimization algorithm, calculate the optimal distance ∆x(t) that sensor node $s_{j}$ needs to move horizontally and the optimal distance ∆y(t) that sensor node $s_{j}$ needs to move vertically, and enter step 5.

Step 5. If the new position to be moved is outside the monitoring area Λ, the moving process will not be carried out, and the wireless sensor node will not move, so go to step 6. Otherwise, move to a new position and go to step 6.

Step 6. If the preset number of cycles is reached, the algorithm ends. Otherwise, go to step 1 and start the next moving process.

## 6. Simulation and Discussion

Python language was used to program the above deployment model. It is assumed that N wireless sensor nodes are randomly deployed in a monitoring area Λ of 100 m × 100 m, with the sensing radius of all nodes $R_{s}$ = 10 m, sensing error range $R_{e}$ = 2 m, and communication radius $R_{c}=3R_{s}=30\;m$. m=100; n=100.

The parameters of the VF algorithm are as follows: $k_{1}=1,\alpha_{1}=2,F_{th}=1,Max\mathrm{Step}=5\;m$. The parameters of this algorithm are as follows: sensing probability parameters, β=2,λ=0.2; gain coefficients, k=2,α=2; VF threshold, $F_{th}=1$; minimum sensing probability allowed by a wireless sensor network, $P_{\mu }=1$; maximum moving distance allowed by a sensor, MaxStep=5 m; sensing radius, R=16 m. The parameters related to the optimization of the particle swarm are set as follows: acceleration factors of the particle swarm, $c_{1}=c_{2}=c_{3}=1$; maximum number of iterations, MaxNumber=300; velocity interval of particles $\left[v_{min},v_{max}\right]=0.2\times \left[0,50\right]=\left[0,10\right]$.

To prevent wireless sensor nodes from moving out of the monitoring area Λ, if the new position to be moved to is located at the edge 5 m wide inside the monitoring area, the nodes will not move and will remain in the original position.

### 6.1. The Deployment Results

The fixed position of the sensor node after the initial random arrangement is shown in Figure 3a. Figure 3b shows the deployment of sensor nodes using the VF algorithm. The hot attraction of fixed nodes to ordinary nodes limits the movement of nodes, resulting in a local accumulation of nodes. Figure 3c shows the optimized deployment effect of particle swarm optimization. The local aggregation phenomenon has been alleviated, and the average coverage rate of the network has reached 83.62. Figure 3d shows the sensor node deployment optimized by the proposed VF-PSO-DS strategy. The blind area of regional coverage is reduced, and the local aggregation phenomenon almost disappears. This shows that the aggregation of the two algorithms plays an effective role in the uniform coverage of the network, and the effective coverage of the optimized network reaches 89.38%.

To further verify the performance of the three algorithms, the convergence rates were compared through experiments, as shown in Figure 4. The original VF strategy does not take moving distance optimization into account, and most work determines the moving distance based on the amount of the virtual force (see [36]). As a result, the VF algorithm provides the lowest coverage rate. In addition, the PSO algorithm does not account for virtual force, and the convergence time is substantially slower; the final optimization result, however, is gratifying.

One hundred independent simulation experiments were conducted. The average effective coverage, average calculation time, and average iteration times of the three algorithms are shown in Table 2. Experimental results show that the proposed VF-PSO-DS strategy can quickly and effectively optimize the deployment of wireless sensor networks, and its calculation time is 85.1% and 41.6% of that of the VF algorithm and particle swarm optimization algorithm, respectively, while the effective coverage of the network increases by 19.34% and 3.37% respectively. Compared with the VF algorithm and particle swarm optimization algorithm, the algorithm not only has a better deployment optimization effect for wireless sensor networks, but also has a faster convergence speed and requires less calculation time.

### 6.2. The Adaptability Tests

To verify the adaptability of the proposed VF-PSO-DS strategy to hot spots and obstacles, it is assumed that the area to be measured contains one hot spot and several obstacles, and the wireless sensor network contains 16 fixed sensor nodes and 80 mobile sensor nodes, while other network parameters and algorithm parameters remain unchanged. The initial deployment of the network is shown in Figure 5a,b. Based on the measurement requirements of different areas in the deployment of the network, the network can effectively avoid obstacles, and the density of sensor nodes in hot spots can be improved. Thus, the VF-PSO-DS strategy can meet the needs in a complex environment, and the algorithm is robust.

The stability of the deployment strategy was tested for the case of the number of sensor nodes increasing. At each of the values of N=300,350,400,450,500, the simulation was carried out 20 times, and the results were averaged. From Table 3 and Table 4, it can be concluded that with the increase in the number of sensor nodes, the coverage of the monitoring area increases, and the moving distance of the nodes also increases. Because the VF algorithm does not control the position of nodes’ movement in the dense nodes, the moving distance is too large, the moving effect is not good, and it is less helpful to improve the coverage rate.

## 7. Conclusions

In this paper, a sensing probability model of nodes is suggested and the sensing probability is aggregated by D-S evidence. Because the traditional D-S evidence theory is inconsistent with the actual situation and the result is invalid when there is a large evidence conflict, the priority factor is introduced to reassign the sensing probability in the part where the evidence conflict occurs. A virtual force-directed particle swarm optimization algorithm based on D-S evidence theory (VF-PSO-DS) for node deployment is proposed. In this approach, the effective coverage rate of the network is taken as the optimization goal, the global optimal scheme is searched for by particle swarm optimization, and the virtual force fine-tuned by sensing probability given by D-S evidence is employed to guide the particle evolution and speed up the convergence. Experimental results show that the virtual force-directed particle swarm optimization approach can quickly and effectively achieve global optimization. Compared to the virtual force algorithm and the particle swarm optimization algorithm, the virtual force-directed particle swarm optimization approach takes less time, converges faster, and increases the sensing coverage of wireless sensor networks.

## Figures and Tables

**Figure 1 entropy-24-01637-f001:**
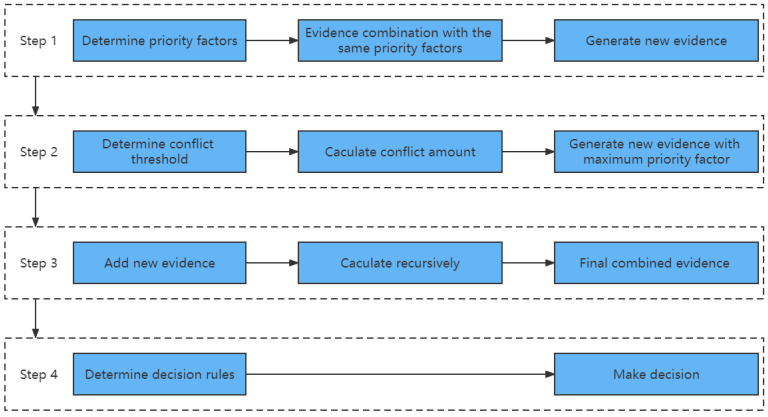
The process of evidence aggregation with priority factors.

**Figure 2 entropy-24-01637-f002:**
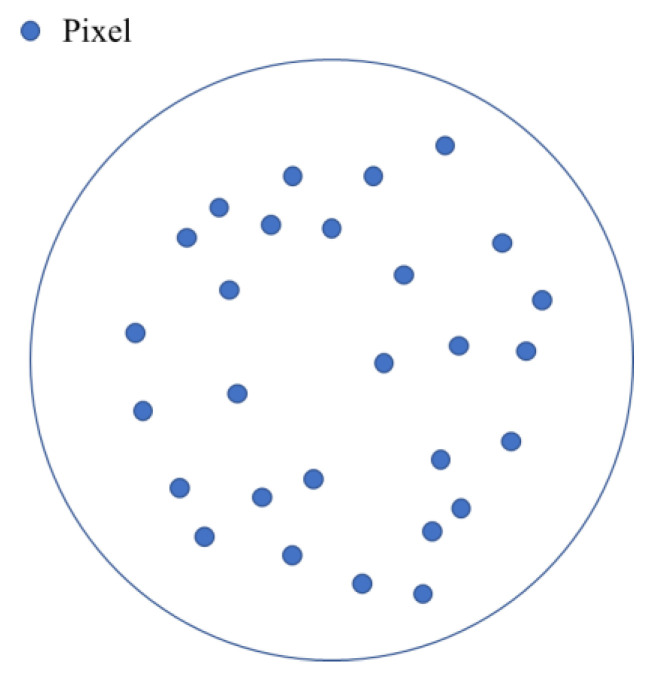
A possible diagram of pixels.

**Figure 3 entropy-24-01637-f003:**
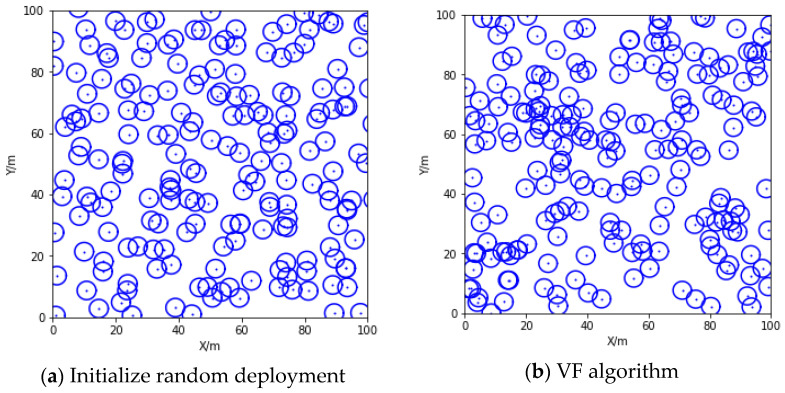
Network coverage using various algorithms.

**Figure 4 entropy-24-01637-f004:**
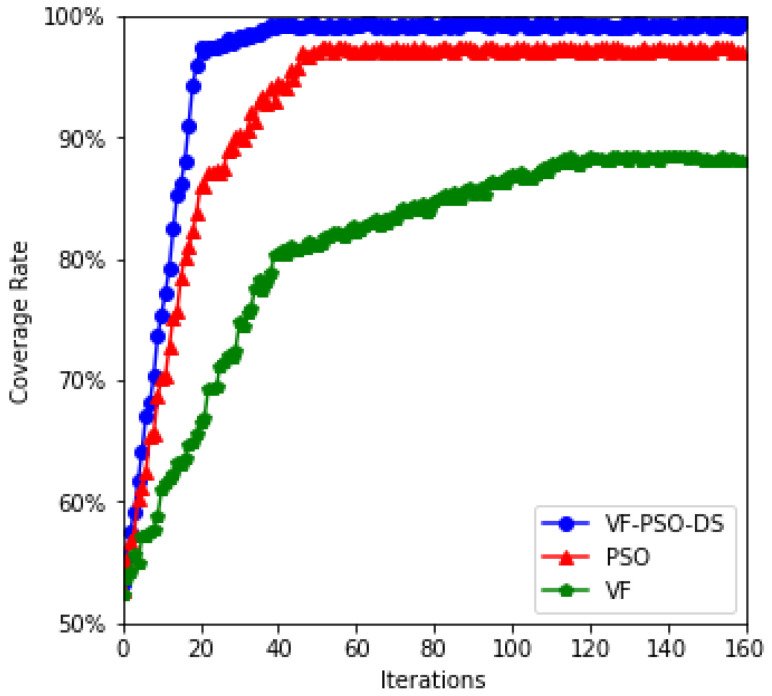
Convergence curves of various algorithms.

**Figure 5 entropy-24-01637-f005:**
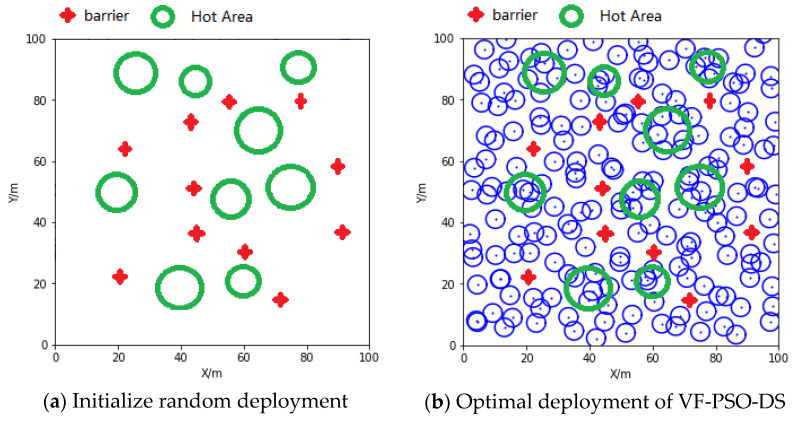
(**a**) Initial network layout with hot spots and obstacles; (**b**) the network layout optimized by VF-PSO-DS approach with hot spots and obstacles.

**Table 1 entropy-24-01637-t001:** Classification of related work by research constraints.

Schemes	Constraints				
	Energy	QoS	Network Lifetime	Network Connectivity	Sensing Coverage
[6]				Y	Y
[7]			Y	Y	Y
[8]			Y		Y
[9]				Y	Y
[10]		Y			Y
[11]	Y				Y
[12]		Y			Y
[13]	Y				Y
[14]					Y

**Table 2 entropy-24-01637-t002:** Statistics of 100 independent optimizations.

	VF	PSO	VF-PSO-DS
Effective coverage (%)	79.29	95.26	98.63
Calculation time	137.43	280.95	116.91
Iterations	161.48	140.33	40.59

**Table 3 entropy-24-01637-t003:** Network coverage using various algorithms when the number of nodes increases.

Number of Sensors	Initial Random Deployment	VF	VF-PSO-DS
300	0.4123	0.5285	0.5524
350	0.4622	0.5551	0.5919
400	0.5275	0.6016	0.6693
450	0.5719	0.6582	0.6916
500	0.6236	0.6923	0.7372

**Table 4 entropy-24-01637-t004:** Total moving distances of nodes using various algorithms when the number of nodes increases.

Number of Sensors	Moving Distance of VF	Moving Distance of VF-PSO-DS
300	314.16 m	256.19 m
350	426.82 m	358.27 m
400	517.65 m	399.23 m
450	578.19 m	451.96 m
500	609.87 m	512.25 m

## Data Availability

Not applicable.

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
