# Peer review of "Node Deployment Optimization for Wireless Sensor Networks Based on Virtual Force-Directed Particle Swarm Optimization Algorithm and Evidence Theory"

_entropy, 2022, doi:10.3390/e24111637_

Round 1

Reviewer 1 Report

The authors report on a node deployment optimization of wireless sensor network based on virtual force directed particle swarm algorithm and D-S evidence theory. The method is demonstrated by computer simulation, and convergence graphics are shown.

            The subject is interesting. However, similar work on virtual force directed particle swarm optimization has already been published by other authors: Xue Wang, Sheng Wang, and Daowei Bi published an article entitled “Virtual Force-Directed Particle Swarm Optimization for Dynamic Deployment in Wireless Sensor Networks” in Advanced Intelligent Computing Theories and Applications (2007). Unfortunately, this work was not cited in the present manuscript.

            If the authors want to publish their work on virtual force directed particle swarm optimization, then they should cite the work of Wang et al. Furthermore, the authors should explain what kind of improvements they have achieved in comparison to Wang et al.

            The authors should also check Figure 3: According to the text, the virtual force directed particle swarm algorithm should have the best convergence rate, but in Figure 3, it has the least.

Reviewer 2 Report

1) What is "D-S evidence"? This must be defined/explained before first use in abstract.

2) "Many heuristic algorithms, including the genetic algorithm, the fish optimization program, the ant colony algorithm, and the particle swarm optimization algorithm, are 38 employed to optimize the placement of sensor nodes." Provide reference for each of the mentioned algorithms. 

3) I believe D-S evidence theory is not the contribution of this manuscript. However, I do not see a single reference citing in Section 3.1, where that theory is introduced. 

4) It is not clear how one can measure/determine the probability of a pixel being sensed by a sensor (i.e., p(d_i) in Eq. (17))

5) What is the final optimization problem? The final problems should be presented in the standard way: cost function and constraints. 

6) I don't understand Figure 3. This figure compares different optimization algorithms. If the optimization problem is the same, all solvers should converge to a single optimal point. If they don't, it means that the designing parameters are not well-tuned. I am not sure such a comparison makes sense and shows the advantages of the deployed optimization algorithm. 

7) Literature review is very weak. Optimizing sensor networks have been studied in many recent publications that could be used to enhance the literature review. For instance, consider discussing the following publications:

[R1]. "Optimizing Sensor Network Coverage and Regional Connectivity in Industrial IoT Systems", 2017. [http://doi.org/10.1109/JSYST.2015.2443045]

[R2]. "A Distributed Method for Linear Programming Problems With Box Constraints and Time-Varying Inequalities", 2019 [http://doi.org/10.1109/LCSYS.2018.2889963]

[R3]. "Energy Optimization in Cluster-Based Routing Protocols for Large-Area Wireless Sensor Networks", 2019. [https://doi.org/10.3390/sym11010037]

Round 2

Reviewer 2 Report

No further comment.